# A Vehicle–Bridge Interaction Element: Implementation in ABAQUS and Verification

Yufeng Dong [1], Wenyang Zhang [2], Anoosh Shamsabadi [3], Li Shi [4,*] and Ertugrul Taciroglu [1]

1   Department of Civil and Environmental Engineering, University of California, Los Angeles, CA 90095, USA; yufengdo@g.ucla.edu (Y.D.); etacir@ucla.edu (E.T.)
2   Texas Advanced Computing Center, University of Texas at Austin, Austin, TX 78758, USA; wzhang@tacc.utexas.edu
3   California High-Speed Rail Authority, 770 L Street, Sacramento, CA 95814, USA; anoosh_shams@yahoo.com
4   College of Civil Engineering, Zhejiang University of Technology, Liuhe Rd. 288, Xihu District, Hangzhou 310023, China
*   Correspondence: lishi@zjut.edu.cn

**Abstract:** Vibration analysis of bridges induced by train loads is a crucial aspect of railway design, particularly considering the complexity of vehicle components such as bogie-suspension systems. Consequently, railway engineers have endeavored to improve the computational efficiency and applicability of train models using the finite-element method. This paper introduces a toolbox implemented in ABAQUS through a user-defined element (UEL) subroutine, which incorporates the vehicle–bridge interaction (VBI) element theory. This toolbox effectively handles diverse vehicle–bridge interaction systems. In the proposed theory, the wheel-track contact force is derived based on the bridge response, eliminating the need for an iterative process and significantly reducing computational workload compared to classical physics-based analysis. The presented approach is validated through a moving sprung mass model and a moving rigid bar model. Furthermore, a case study is conducted on a three-dimensional finite-element model of a high-speed railway bridge in China, based on a design sketch, to showcase the capabilities of the developed scheme. The study demonstrates the practical application of the proposed methodology in analyzing vehicle–bridge structures with high complexity.

**Keywords:** vehicle–bridge interaction (VBI); high-speed railway; ballast less slab track; finite-element method; ABAQUS; user-defined element (UEL)

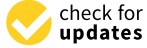



## 1. Introduction

The study of railway–bridge systems under train loads has evolved from simple to complex models [1–5]. With the advancements in train–track coupled dynamic systems worldwide, particular attention has been directed towards the analysis of vehicle–bridge interaction (VBI), especially when the railway is subjected to moving loads from multiple wheels [6,7]. The earliest VBI models focused on a simple support beam subjected to a moving point load. Mathews [8,9] and Trochanis et al. [10] first conducted dynamic analyses of beams resting on elastic foundations subjected to moving loads. Ono and Yamada [11] developed the analytical solution of track vibration using an assumed continuous support system, taking into account the ballast mass effect. Jezequel [12] analyzed the infinite periodic Euler–Bernoulli beam system under moving vehicle loads, regarded as concentrated forces with uniform traversal speeds. With the application of modal superposition, Weaver et al. [13] solved the equations of motion for simply supported beams subjected to moving loads in the time domain. Warburton [14] solved this equation analytically and established that the deflection of beams would reach a maximum under moving loads at a certain speed. Further research has focused on the interaction between the vehicle and the beam. Knothe and Grassie [15–17] explored track dynamics and train–track interaction

in time and frequency domains. Zheng and Fan [18] studied the dynamic instabilities of train–track coupled systems. Chen et al. [19] analyzed the near-fault velocity pulse effect on a vehicle–bridge structure induced by seismic motions. Extensive studies conducted by Zhang et al. and Antolin et al. [20,21] provide a nonlinear wheel-track contact force scheme. Song et al. [22] explored the dynamic behavior of the pantograph–catenary system under vehicle–track excitation and emphasized the importance of spatial effects to better understand and improve the reliability and performance of high-speed railway systems. The authors introduced a novel approach to consider the spatial vibration of the carbody in the pantograph–catenary interaction by developing a spatial contact model between the contact wire and the pantograph strip. Carnicero et al. [23] presented an advanced pantograph–catenary–vehicle–track model, which analyzes the vertical dynamics of the complete system. The model developed by the authors is able to evaluate the displacements and the contact force generated in the catenary–pantograph as well as the wheel-track interactions. In addition, the study also focused on the possible influence of track irregularities on the catenary–pantograph dynamic interaction. Cai et al. analyzed poroelastic ground vibration based on Biot's theory [24–26], while Lei conducted systematic studies on track dynamics [27,28]. Lei introduced a simplified vehicle model with a primary suspension system on a single wheel and a complete car model with primary and secondary suspension systems on four wheels, involving 10 degrees of freedom. Lei [29] subsequently developed the equations of motion for the vehicle–track coupling system, considering the effects of track irregularities and abrupt changes in track stiffness.

Moreover, the critical velocity is also widely studied in the vehicle–bridge interaction field. Kiani and Mehri [30] conducted a dynamic analysis of nanotube structures under the influence of a moving nanoparticle, utilizing the nonlocal continuum theory. They modeled the nanotube structure as an equivalent continuum structure (ECS) based on nonlocal Euler–Bernoulli, Timoshenko, and higher-order beam theories. The study established nondimensional equations of motion for the nonlocal beams, leading to analytical solutions and explicit derivation of critical velocities. Kiani [31] also proposed mathematical models for the vibration of double-walled carbon nanotubes (DWCNTs) subjected to a moving nanoparticle, employing nonlocal classical and shear deformable beam theories. The study assumed the van der Waals (vdW) interaction force between adjacent CNTs as a linear function of the deflection fields of the innermost and outermost CNTs. Mathematical modeling and analytical solutions were conducted, expressing critical velocities associated with nonlocal beam theories in terms of small-scale effect parameters, DWCNT geometry, and material properties. The study's conclusions revealed that the disappearance of the vdW interaction force and the small-scale effect parameter from a slender DWCNT led to predicted critical velocities approaching those of the innermost carbon nanotube based on classical continuum theory. These findings underscored the significance of considering nonlocal effects and interaction forces when analyzing nanotube structures under the influence of moving nanoparticles.

The Euler–Bernoulli beam theory assumes that the cross-sections remain perpendicular to the beam axis during deformation, indicating that the shear strain is negligible. However, in reality, shear deformation can significantly affect the behavior of beams, especially when dealing with slender or highly flexible structures. In such cases, higher-order theories such as the Timoshenko beam theory or shear deformable beam theories should be taken into consideration, which accounts for shear deformation effects more accurately. These higher-order theories provide more accurate predictions for beams with varying slenderness ratios and are better suited for analyzing structures where shear deformation plays a significant role. Lee [32] investigated the analytical solution of the dynamic response of a Timoshenko beam resting on a Winkler-type elastic foundation subjected to a moving concentrated mass, with the possible separation of the moving mass from the beam considered. Yavari et al. [33] proposed a discrete element analysis of the dynamic response of Timoshenko beams under moving mass. Lou et al. [34] conducted the finite-element analysis for a Timoshenko beam subjected to a moving mass. Esen [35] analyzed the dynamic response of a functionally

graded Timoshenko beam on two-parameter elastic foundations due to a variable velocity moving mass. Kiani et al. [36,37] conducted parametric studies to predict the capabilities of classical and shear deformable beam models excited by a moving mass. The beam slenderness, mass weight, and velocity of the moving mass are introduced as well as various boundary conditions. They defined the critical beam slenderness in which, for slenderness lower than it, the application of Euler–Bernoulli or even Timoshenko beam theories would underestimate the real dynamic response of the system. Kiani and Wang [38,39] also extended the beam theory at the nanoscale.

Previous studies have employed various numerical methods such as the Newmark and Runge–Kutta schemes to solve the governing equations for vehicle–bridge interaction problems [40–42]. These methods often involve an iterative procedure within the time-stepping process, as the vehicle and bridge systems have coupled equations of motion. However, the iterative method may suffer a low convergence rate, especially when dealing with complex systems with a large number of wheels. To remedy this, Yang and Wu [43] developed a versatile element method called the VBI element, capable of solving systems with multiple interaction forces by formulating them based on the degrees of freedom (DOF) of the wheels. In this approach, the sub-matrices of the beam element bearing wheels are updated at each time step during analysis. Nevertheless, the challenge lies in efficiently and effectively implementing this approach into commercial finite-element software, which can be crucial when analyzing realistic bridges and high-speed railways that requires a higher fidelity in scaling the numerical models to physical structures. Hence, the primary objective of this paper is to develop a validated toolbox that implements Yang's VBI element theory within ABAQUS [44].

In this study, we have implemented the VBI element theory in ABAQUS using a user-defined element (UEL) subroutine. To validate the accuracy and effectiveness of the VBI element application, we employ two classical models: a single moving sprung mass on a simple support beam and a rigid moving bar on a beam. By comparing the numerical results obtained from the VBI element implementation with analytical solutions, we verify the reliability of the approach. Furthermore, we perform a numerical simulation using a three-dimensional model based on a high-speed railway bridge in China. The purpose of this simulation is to demonstrate the validity of the proposed implementation procedure in realistic structures. Herein, we showcase the capability of the implemented approach to capture the dynamic behavior of complex railway bridges. It is worth noting that the partial validations obtained from the simulations provide further evidence of the effectiveness of the proposed implementation. To facilitate the broader use of this research, we will make the implemented codes publicly available.

## 2. Formulations for Vehicle–Bridge Interaction (VBI) Element Applied in ABAQUS

The VBI element theory focused on establishing the interaction between contact force and degrees of freedom (DOF) on beam elements. The formulations utilized in the ABAQUS user-defined element (UEL) subroutine are derived from the equations of motion for dynamic subsystems as outlined in Yang's theory [43,45,46].

### 2.1. VBI Element Formulation

Two sets of equations of motion are introduced for the vehicle and bridge subsystems. The vehicle's equation of motion is presented in lumped matrix form, with the degrees of freedom (DOF) separated into two parts: carriage and wheels. The force vector is divided into the external force, acting on both the carriage and wheels, and the contact force, which only imposes on wheels, as shown in Equation (1) [43],

$$\begin{bmatrix} [m_{\mathrm{uu}}] & [m_{\mathrm{uw}}] \\ [m_{\mathrm{wu}}] & [m_{\mathrm{ww}}] \end{bmatrix} \left\{ \begin{array}{c} \{\ddot{d}_{\mathrm{u}}\} \\ \{\ddot{d}_{\mathrm{w}}\} \end{array} \right\}_{t+\Delta t} + \begin{bmatrix} [c_{\mathrm{uu}}] & [c_{\mathrm{uw}}] \\ [c_{\mathrm{wu}}] & [c_{\mathrm{ww}}] \end{bmatrix} \left\{ \begin{array}{c} \{\dot{d}_{\mathrm{u}}\} \\ \{\dot{d}_{\mathrm{w}}\} \end{array} \right\}_{t+\Delta t}$$
$$+ \begin{bmatrix} [k_{\mathrm{uu}}] & [k_{\mathrm{uw}}] \\ [k_{\mathrm{wu}}] & [k_{\mathrm{ww}}] \end{bmatrix} \left\{ \begin{array}{c} \{d_{\mathrm{u}}\} \\ \{d_{\mathrm{w}}\} \end{array} \right\}_{t+\Delta t} = \left\{ \begin{array}{c} \{f_{\mathrm{ue}}\} \\ \{f_{\mathrm{we}}\} \end{array} \right\}_{t+\Delta t} + \begin{bmatrix} [l_{\mathrm{u}}] \\ [l_{\mathrm{w}}] \end{bmatrix} \{f_{\mathrm{c}}\}_{t+\Delta t} \quad (1)$$

where $\{d_{\mathrm{u}}\}$ and $\{d_{\mathrm{w}}\}$ denote the DOF of carriage and wheels, respectively; $\{f_{\mathrm{ue}}\}$ and $\{f_{\mathrm{we}}\}$ represent the external forces acting on the carriage and wheels; and $\{f_{\mathrm{c}}\}$ is the contact force vector that is distributed to DOF of carriage and wheels through matrices $[l_{\mathrm{u}}]$ and $[l_{\mathrm{w}}]$. The interaction force is given in terms of DOF for wheels. Derailment is neglected in the following analysis for simplicity.

The displacement, velocity, and acceleration at the time step $t + \Delta t$ are obtained using Newmark's time-stepping scheme. These quantities are used to express the contact force between the wheels and the track in terms of the wheels' degrees of freedom (DOF). The governing equation of the carriage is expanded to achieve this. As this paper does not consider derailment, the DOF of the wheels is substituted with the DOF of the contact points between the wheels and the rail.

According to beam element theory in finite-element analysis [47], the degrees of freedom (DOF) for the contact points can be expressed in terms of the DOF for nodes of the beam element using shape function interpolation, as illustrated in Equations (9)–(12). At the time step $t + \Delta t$, we suppose there are $n$ wheels on the rail. In this case, the beam element subjected to wheel load will be transferred to the VBI element. The $i$th VBI element equation of motion is provided by:

$$[m_{\mathrm{b}}]\{\ddot{d}_{\mathrm{b}i}\}_{t+\Delta t} + [c_{\mathrm{b}i}]\{\dot{d}_{\mathrm{b}i}\}_{t+\Delta t} + [k_{\mathrm{b}i}]\{d_{\mathrm{b}i}\}_{t+\Delta t} = \{f_{\mathrm{b}i}\}_{t+\Delta t} - \{f_{\mathrm{b}ci}\}_{t+\Delta t} \tag{2}$$

where $[m_{\mathrm{b}i}]$, $[c_{\mathrm{b}i}]$ and $[k_{\mathrm{b}i}]$ represent the mass, damping, and stiffness matrices of the ith element of the rail, respectively; $\{d_{\mathrm{b}i}\}$ is the nodal displacement vector; $\{f_{\mathrm{b}i}\}$ is the external force vector; and $\{f_{\mathrm{b}ci}\}$ is the vector of equivalent nodal forces related to the ith factor of the contact force vector $\{f_{\mathrm{c}}\}$:

$$\{f_{\mathrm{b}ci}\}_{t+\Delta t} = \{N_{ci}^{\mathrm{v}}\}f_{ci,t+\Delta t} \tag{3}$$

where $N_i^v$ denotes the shape function of the ith bridge element, which is represented by Hermitian cubic polynomials.

By applying $\{f_{ci}\}$ into $\{f_{\mathrm{b}ci}\}$ and expressing $\{f_{ci}\}$ in terms of $\{d_{\mathrm{b}i}\}$ [43], we obtain the updated equation of motion for the $i$th VBI element:

$$\begin{aligned}&[m_{\mathrm{b}i}]\{\ddot{d}_{\mathrm{b}i}\}_{t+\Delta t} + [c_{\mathrm{b}i}]\{\dot{d}_{\mathrm{b}i}\}_{t+\Delta t} + [k_{\mathrm{b}i}]\{d_{\mathrm{b}i}\}_{t+\Delta t}\\ &= \{f_{\mathrm{b}i}\}_{t+\Delta t} - \sum_{j=1}^{n}\left(\left[m_{cij}^*\right]\{\ddot{d}_{\mathrm{b}j}\} + \left[c_{cij}^*\right]\{\dot{d}_{\mathrm{b}j}\} + \left[k_{cij}^*\right]\{d_{\mathrm{b}j}\}\right) - \{p_{ci}^*\}_{t+\Delta t} - \{q_{ci}^*\}\end{aligned} \tag{4}$$

where

$$\left[m_{cij}^*\right] = \{N_{ci}^{\mathrm{v}}\}m_{cij}\langle N_c^{\mathrm{v}}\rangle \tag{5}$$

$$\left[c_{cij}^*\right] = \{N_{ci}^v\}c_{cij}\langle N_{cj}^v\rangle \tag{6}$$

$$\left[k_{cij}^*\right] = \{N_{ci}^{\mathrm{v}}\}k_{cij}\langle N_{cj}^{\mathrm{v}}\rangle \tag{7}$$

and the equivalent force vectors are

$$\{p_{ci}^*\}_{t+\Delta t} = \{N_{ci}^{\mathrm{v}}\}p_{ci,t+\Delta t}, \quad \{q_{ci}^*\}_t = \{N_{ci}^{\mathrm{v}}\}q_{ci,t} \tag{8}$$

where the contact matrices $[m_{\mathrm{c}}]$, $[c_{\mathrm{c}}]$ and $[k_{\mathrm{c}}]$ and the load vectors $\{p_{\mathrm{c}}\}$ and $\{q_{\mathrm{c}}\}$ are related to the mass, damping, and stiffness matrices and the contact force and external force vectors for the vehicle system.

The details of the algebra process are presented in [43]. Based on the above algorithm, the equation of motion for a VBI element can be solved directly by updating its stiffness, damping, and mass matrices without iteration, enabling ABAQUS to realize it through a user-element subroutine (UEL).

### 2.2. Implementation in ABAQUS through UEL Subroutine

To implement the VBI element theory in ABAQUS, we propose a user-defined element (UEL) subroutine, in which 2-node beam elements are applied. The Hermite interpolation functions are based on the shape functions of the beam element:

$$N_1 = 1 - 3\left(\frac{x}{l_e}\right)^2 + 2\left(\frac{x}{l_e}\right)^3 \tag{9}$$

$$N_2 = l_e\left[-\left(\frac{x}{l_e}\right) + 2\left(\frac{x}{l_e}\right)^2 - \left(\frac{x}{l_e}\right)^3\right] \tag{10}$$

$$N_3 = 3\left(\frac{x}{l_e}\right)^2 - 2\left(\frac{x}{l_e}\right)^3 \tag{11}$$

$$N_4 = l_e\left[\left(\frac{x}{l_e}\right)^2 - \left(\frac{x}{l_e}\right)^3\right] \tag{12}$$

where $x$ is the element local coordinate, and $l_e$ represents element length. It should be noticed that the shape functions applied in this work are based on the Euler–Bernoulli beam theory, assuming that the cross-sections remain perpendicular to the beam axis during deformation, implying that the shear strain is negligible. As a result, the Euler–Bernoulli beam theory tends to provide accurate results for beams with low slenderness ratios and negligible shear effects. For beams with high slenderness ratios or subjected to significant shear forces, it becomes essential to consider higher-order theories, such as the Timoshenko beam theory or shear deformable beam theories, which account for shear deformation effects more accurately.

The parameters of the Newmark time-stepping method are also defined in [43]. When dealing with the VBI element, we apply a loop to judge which wheel is on the current element. The element will become a VBI element as long as the position of one wheel is within the region of the coordinates:

$$\begin{aligned} x_{\mathrm{n}} &= x_0(i) \\ x_{\mathrm{c}} &= x_{\mathrm{n}} + vt \end{aligned} \tag{13}$$

where $x_n$ represents the initial position of the $i$th wheel, $t$ denotes time, and $x_c$ represents the current position of the $i$th wheel.

Based on Yang's theory [43], we update stiffness and mass matrices through Equations (4)–(8) by moving matrix terms from right to left:

$$[M]\{\ddot{D}\}_{t+\Delta t} + [C]\{\dot{D}\}_{t+\Delta t} + [K]\{D\}_{t+\Delta t} = \{F_{\mathrm{b}}\}_{t+\Delta t} - \{P_{\mathrm{c}}^*\}_{t+\Delta t} - \{Q_{\mathrm{c}}^*\} \tag{14}$$

where

$$\begin{aligned} [M] &= [M_{\mathrm{b}}] + [M_{\mathrm{c}}^*] = \sum[m_{\mathrm{b}i}] + \sum\sum\left[m_{\mathrm{c}ij}^*\right] \\ [C] &= [C_{\mathrm{b}}] + [C_{\mathrm{c}}^*] = \sum[c_{\mathrm{b}i}] + \sum\sum\left[c_{\mathrm{c}ij}^*\right] \\ [K] &= [K_{\mathrm{b}}] + [K_{\mathrm{c}}^*] = \sum[k_{\mathrm{b}i}] + \sum\sum\left[k_{\mathrm{c}ij}^*\right] \\ \{P_{\mathrm{c}}^*\}_{t+\Delta t} &= \sum\{p_{ci}^*\}_{t+\Delta t} \\ \{Q_{\mathrm{c}}^*\}_t &= \sum\{q_{ci}^*\} \end{aligned} \tag{15}$$

In ABAQUS, the default time-stepping approach—namely, the Hilber–Hughes–Taylor (HHT-$\alpha$ method) [44] implicit time integration scheme—is utilized in the analysis. In the UEL subroutine, the effective stiffness matrix ([$AMATRX$]) and the residual vector ({$RHS$}) are adopted to compute the increment of DOF as shown below, which is updated in every step:

$$[AMATRX]\{\Delta u\} = \{RHS\} \tag{16}$$

where [*AMATRX*] can be shown as

$$[AMATRX] = \frac{1}{\beta \Delta t^2}[M] + \frac{(1+\alpha)\gamma}{\beta \Delta t}[C] + (1+\alpha)[K] \tag{17}$$

For the residual vector ({*RHS*}), Yang's theory is applied for updating the state variables (*SVARS*) and {*G*} vector:

$$\{RHS\} = -[M]\{\ddot{u}\}_{t+\Delta t} + (1+\alpha)\{G\}_{t+\Delta t} \quad -\alpha\{G\}_t \tag{18}$$

$$\{G\}_{t+\Delta t} = -[K]\{u\}_{t+\Delta t} - [C]\{\dot{u}\}_{t+\Delta t} + \{f_e\}_{t+\Delta t} - \{q_c\}_t \tag{19}$$

where $\{f_e\}$ represents the external force, usually gravity.

The default value of the numerical damping parameter $\alpha$ in the HHT method is -1/20. The details of the algorithm are included in the codes that are provided as Supplementary Materials. A flow chart of the main structure of the finite-element analysis software ABAQUS for the problem concerned is shown in Figure 1. In the flowchart, $\{d_u\}$ represents the degrees of freedom (DOF) of the carriage, and $\{d_w\}$ represents the DOF of the wheels. Additionally, $\{d_c\}$ corresponds to the DOF of the contact point, and $\{d_b\}$ corresponds to the DOF of the beam element.

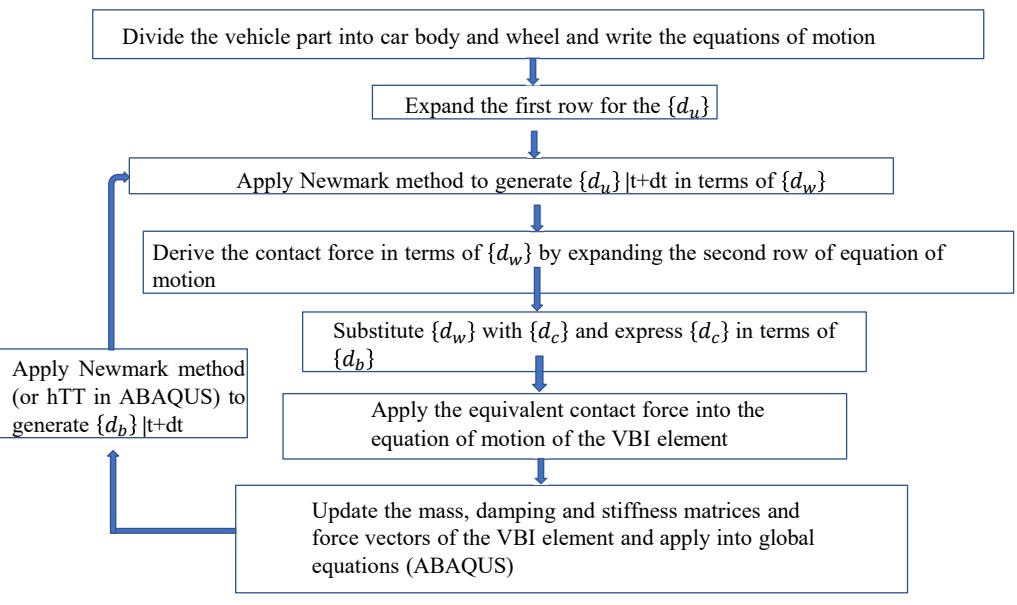

**Figure 1.** The logic flowchart of VBI element theory implementation.

## 3. Verification

In this section, we conduct a verification of the implemented VBI element by comparing its results with analytical solutions. Specifically, we examine the accuracy and applicability of the VBI element in two scenarios: a simple-supported beam subjected to a sprung mass and a suspended rigid beam. By performing this verification, we aim to assess the reliability and suitability of the VBI element in simulating these dynamic situations and ensure its proper functioning in the analysis of vehicle–bridge interactions.

### 3.1. Sprung Mass System

A simply supported beam subjected to a moving sprung mass system [43,45], as illustrated in Figure 2, is analyzed with the application of the VBI element. The parameters are Young's modulus $E = 2.87$ GPa, Poisson's ratio $\nu = 0.2$, the moment of inertia $I = 2.90$ m$^4$, mass per unit length $m = 2303$ kg/m, girder length $L = 25$ m, sprung mass $M_v = 5.75$ ton,

wheel mass $M_w = 0$, and suspension spring constant $k_v$ are considered as an infinite number to simulate an equivalent moving mass, which is consistent with the analytical solution referred in the following. The dashpot $c_v$ is neglected. The vehicle velocity is 27.78 m/s. The beam is divided into 100 2-node elements. The results of the displacement and velocity for the midpoint of the beam are compared with the analytical solution based on the first-mode approximation [48] obtained by MATLAB [49] in Figure 3. The results show a good match between the two methods of realization, demonstrating the availability of the implementation for the VBI element theory in ABAQUS. The vibration of results generated through the VBI element theory is attributed to the higher modes, which are neglected in the analytical solution.

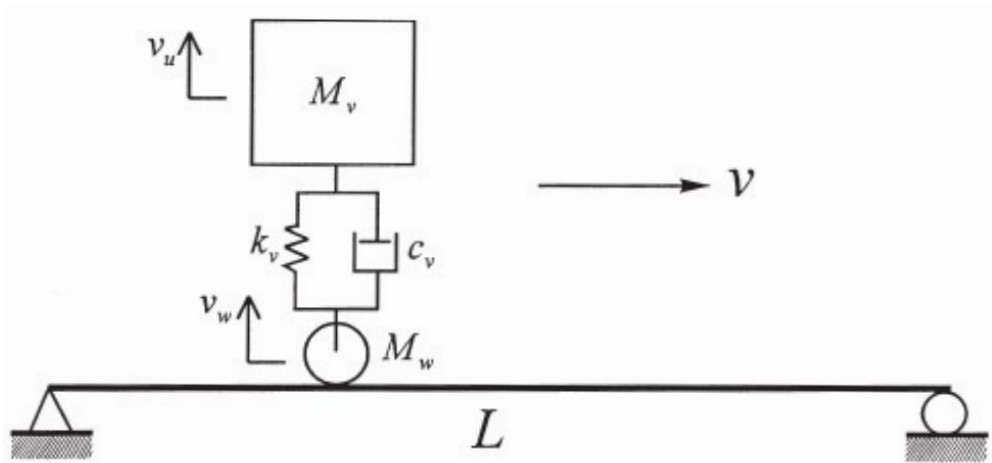

**Figure 2.** Schematic representation of the sprung mass model.

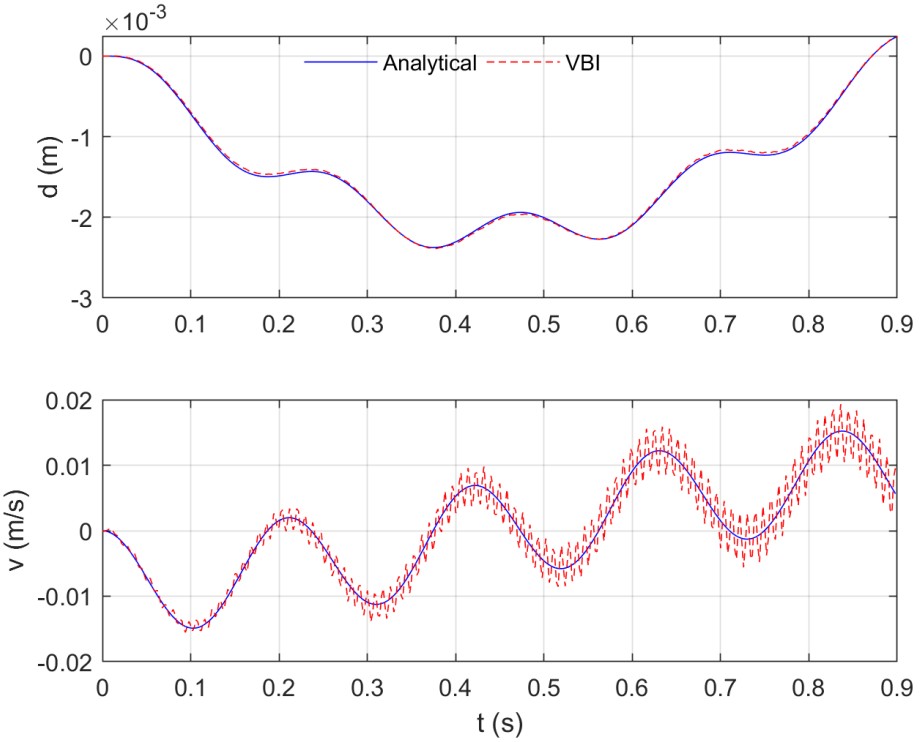

**Figure 3.** Midpoint displacement (**top**) and velocity (**bottom**) of the beam for the sprung mass model.

### 3.2. Moving Suspended Rigid Beam System

The second verification model considers a simply supported beam subjected to a suspended rigid beam supported by two wheels with spring and dashpot systems [43,45],

as shown in Figure 4. The parameters for the model are Young's modulus $E = 2.943$ GPa, Poisson's ratio $\nu = 0.2$, moment of inertia $I = 8.65$ m$^4$, mass per unit length $m = 36$ t/m, beam length $L = 30$ m, $M_v = 540$ ton, wheel masses are zero, rotatory mass $I_v = 13{,}800$ t m$^2$, and suspension spring constant $k_v$ are considered as an infinite value. The dashpot parameter $c_v$ is set as zero. The distance between the two wheels is 17.5 m. The vehicle speed is the same as in the first example. The beam meshes into 60 elements. The results of the displacement and velocity for the midpoint of the beam are compared with the analytical solution based on the first-mode approximation [48] obtained by MATLAB, as illustrated in Figure 5. Again, a good match is obtained for the proposed implementation with the application of the VBI element, indicating the availability of the subroutine realizing the VBI element theory. The vibration in the response with the application of the VBI element theory comes from the higher-order mode contributions, which are neglected in the analytical solution.

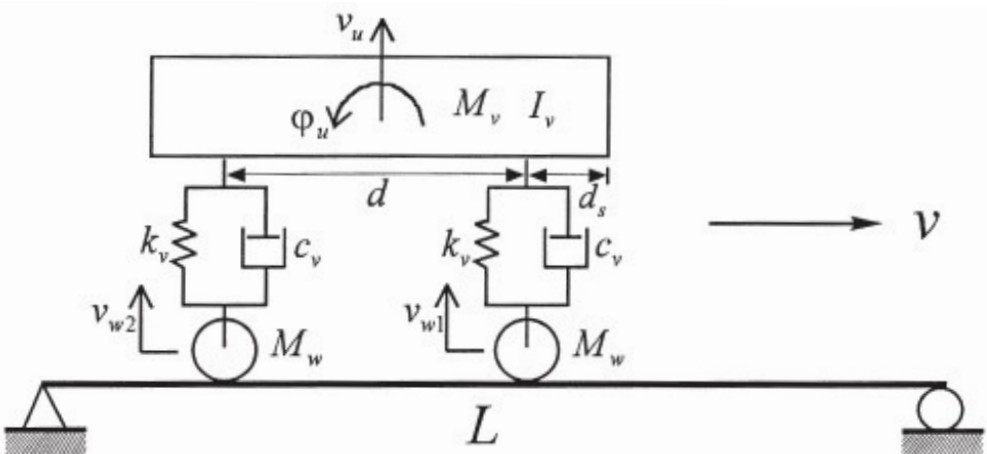

**Figure 4.** The moving suspended rigid beam system.

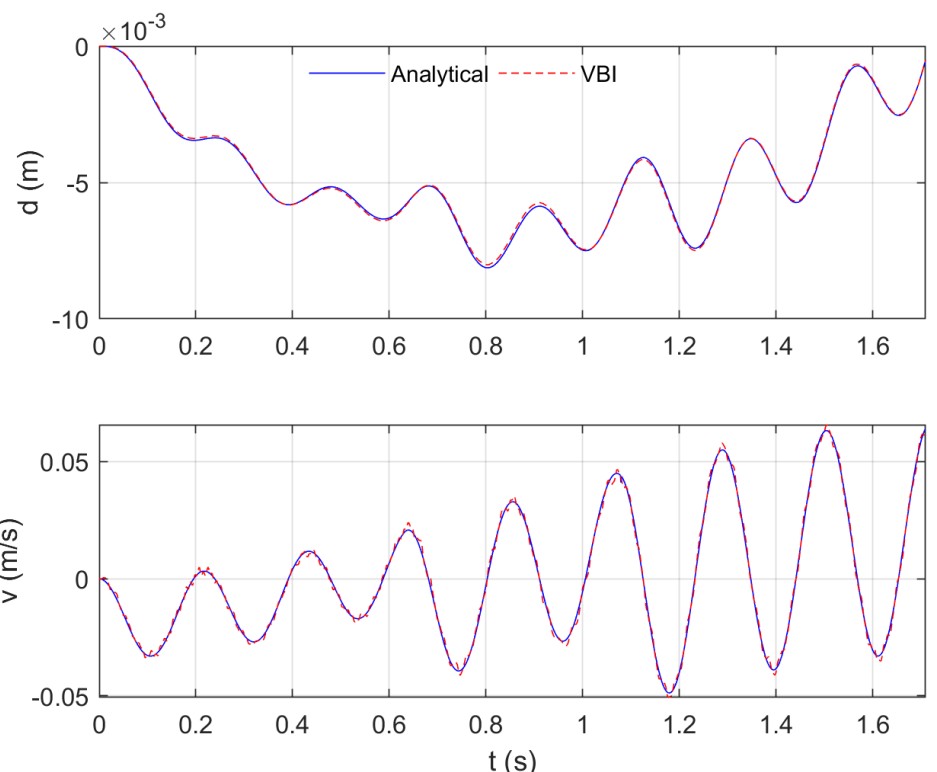

**Figure 5.** Midpoint displacement (**top**) and velocity (**bottom**) of the beam for the moving rigid bar system

## 4. Case Study

Upon the successful verification of the VBI element theory in ABAQUS, we proceeded with the simulation of a high-speed railway bridge. This simulation involved the utilization of the user-element (UEL) VBI element subroutine, which we had implemented. By applying the UEL VBI element subroutine, we aimed to analyze the dynamic behavior of the high-speed railway bridge.

### 4.1. General Information

A three-dimensional model simulation was carried out based on the design sketch of a ballast-less high-speed railway bridge located under the city railway line in southeastern China. The bridge is composed of spans with a length of 34.9 m. The box girder material is C50 concrete, supporting railway tracks. The track comprises the CHN60 rail, the WJ-7B rail fastener, the block-type sleeper, the C40 concrete slab, and the rubber layer. The sleeper and the slab are cast together, and the rubber layer lays between the slab and the girder surface, which attenuates the vibration. The girders are supported by concrete piers of C50 concrete, which rest on the pile-elevated soil foundation. The schematic diagram of the railway bridge is demonstrated Figures 6 and 7. The parameters of materials are illustrated in Table 1. We take the case of the train speed as 60 km/h, and the observation point is A2 for the FE model, which is located in the middle of the second span, as shown in Figure 8. The train load consists of four carriages with four wheels each. The interaction between wheels and rail is as shown in Figure 9. The tracks and bearings are demonstrated in Figures 10 and 11.

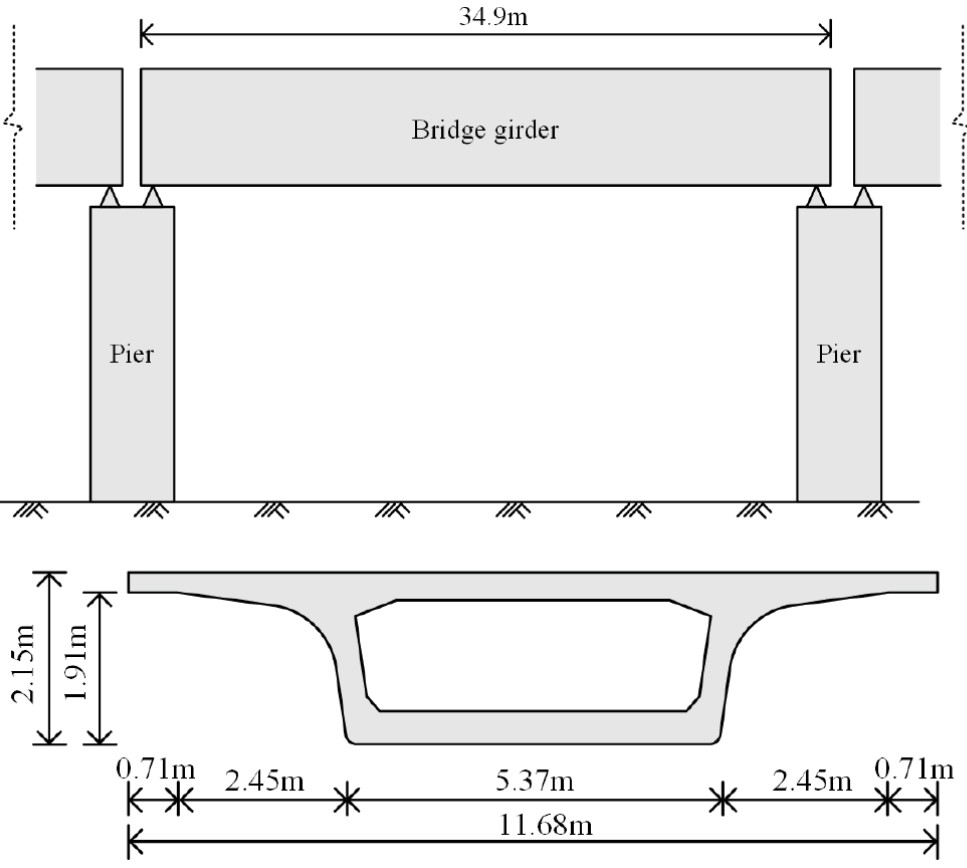

**Figure 6.** Schematic side and front views of the bridge.

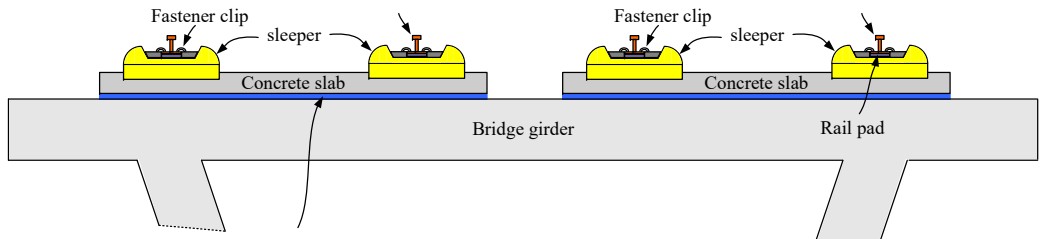

**Figure 7.** Schematic view of the rubber-pad floating slab track.

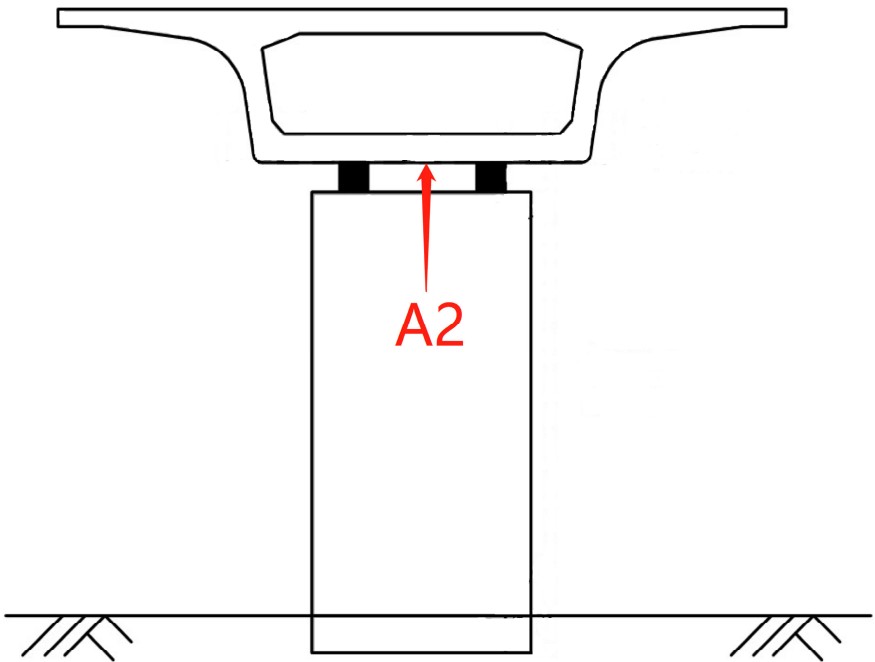

**Figure 8.** Interested node A2.

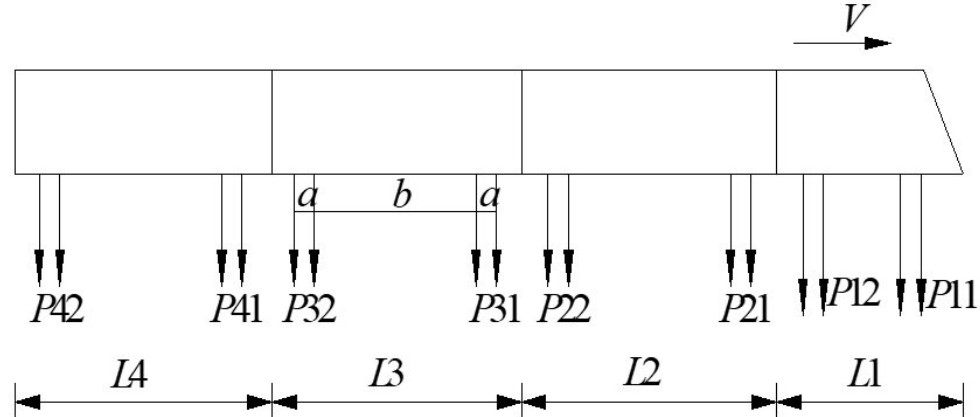

**Figure 9.** Schematic drawing on distribution of axle loads of the test train.

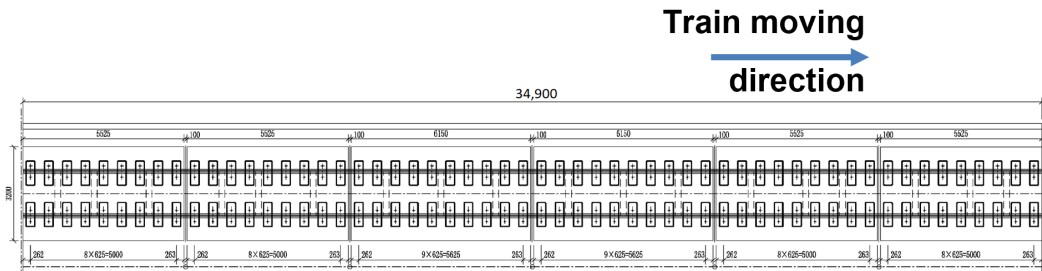

**Figure 10.** Top views of the tracks.

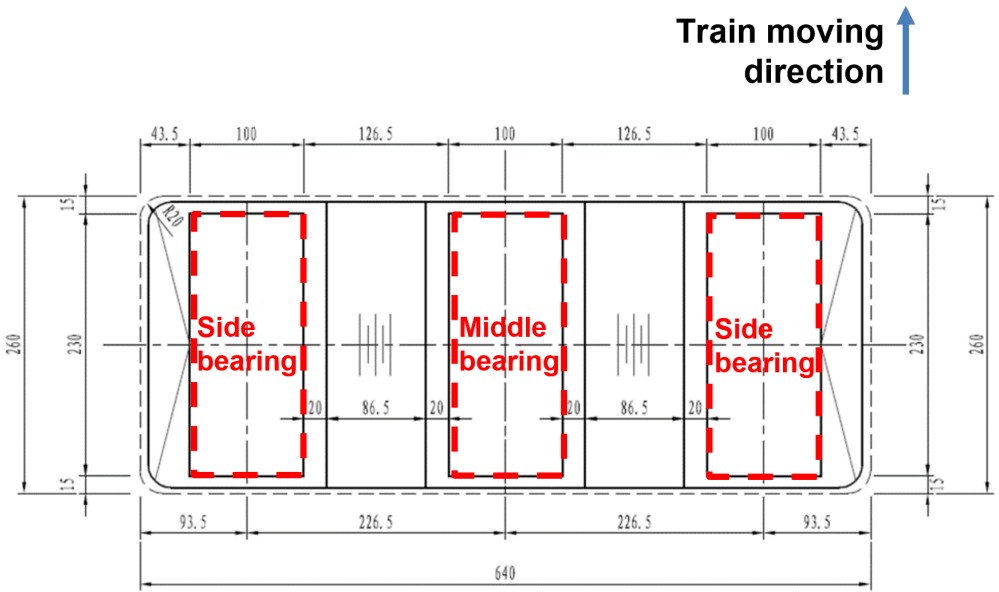

**Figure 11.** Top views of the bearings and piers.

**Table 1.** Model parameters.

| Parts | Young's Modulus (Pa) | Poisson's Ratio | Mass Density |
|---|---|---|---|
| Rail | $2 \times 10^{11}$ | 0.26 | 7850 |
| Rail pad | $3.25 \times 10^{10}$ | 0.2 | 2440 |
| Rail pad reduction layer | $5.18 \times 10^{9}$ | 0.48 | 1700 |
| Box girder | $1.98 \times 10^{10}$ | 0.2 | 2500 |
| Bearing | $2 \times 10^{11}$ | 0.26 | 7850 |
| Pier | $3.15 \times 10^{10}$ | 0.2 | 2420 |

*4.2. Model Introduction and Numerical Results*

In our FE model in ABAQUS, we create the simplified railway bridge according to the design document. The train load is modeled as a suspended vehicle carriage (rigid mass supported by a combination of stiffness and damper) resting on a rigid wheel mass. Moreover, the irregularities between the wheel and the road/rail are considered following the theory [45,46]. The irregularity factors are assigned to external force terms. However, contact stiffness is not included. In other words, the wheels are directly in contact with the road. The decks, bearings, and piers are modeled as solid elements(C3D8). The beam element is adopted for the rail part with the vertical displacement and rotational DOF, while the other four are fixed. The rail pad is modeled using shell elements. We use spring and dashpot systems to connect nodes on the rail and rail pad instances. Since the track lines are over 1000 m in reality, it should be considered that the model constructed in ABAQUS only covers three spans for the convenience of applying the VBI element technique, which

will be stated in the data interpretation section. Here, we define the x, z, and y axles as the longitudinal, transverse, and vertical directions of the bridge. Only vertical vibrations, i.e., along the y axle, are recorded and interpreted in this paper. The 3D FE model in ABAQUS is illustrated in Figure 12. Table 2 summarizes the input parameters of the VBI element.

**Table 2.** Model parameters applied in the UEL subroutine.

| Parameter | Symbol | Value |
|---|---|---|
| Mass density of beam | $\rho_b$ | 7850 kg/m$^3$ |
| Section area of beam | $A_b$ | 0.00764 m$^2$ |
| Bending rigidity of beam about y axle | $EI_{by}$ | $6.11 \times 10^{-8}$ N m$^2$ |
| Bending rigidity of beam about z axle | $EI_{bz}$ | $6.11 \times 10^{-8}$ N m$^2$ |
| Mass of vehicle carriage | $M_v$ | 10,000–11,700 kg |
| Mass of vehicle wheel | $M_w$ | 675 kg |
| Stiffness of vehicle suspension | $k_{sus}$ | $1.4 \times 10^6$ N/m |
| Damping of vehicle suspension | $c_{sus}$ | $3 \times 10^4$ N s/m |
| Initial positions for wheels | $x_0$ | −77.45–11 m |
| Train velocity | $v$ | 60 km/h |
| Wheel number | $n_c$ | 16 |

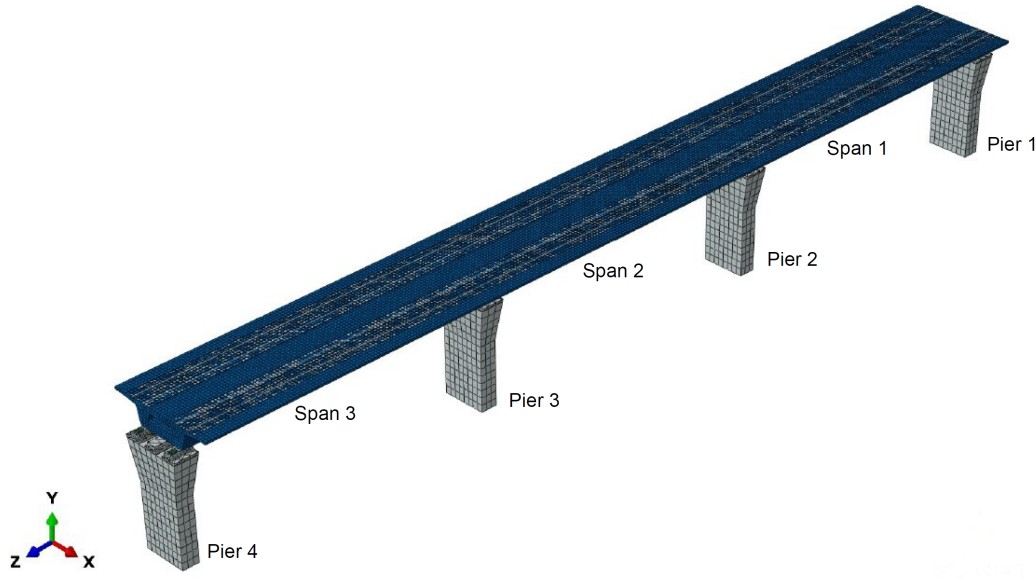

**Figure 12.** The 3D FE model of railway bridge in ABAQUS.

The soil dynamics are neglected in the simulation, meaning the piles and soil basements are not built since this study focuses on implementing the VBI element theory in ABAQUS. We presented the nodal velocity at node A2 for the time history of the train passing through and 10 s free decay after the train left. The results are shown in Figures 13–15.

Figure 13 presents the numerical results in the time domain for velocities in node A2. It also demonstrates the data under a high-frequency filter and low-frequency filter. To better evaluate the accuracy of the simulation, we conducted the simulation for ten more seconds after the train load left to capture the structure's character.

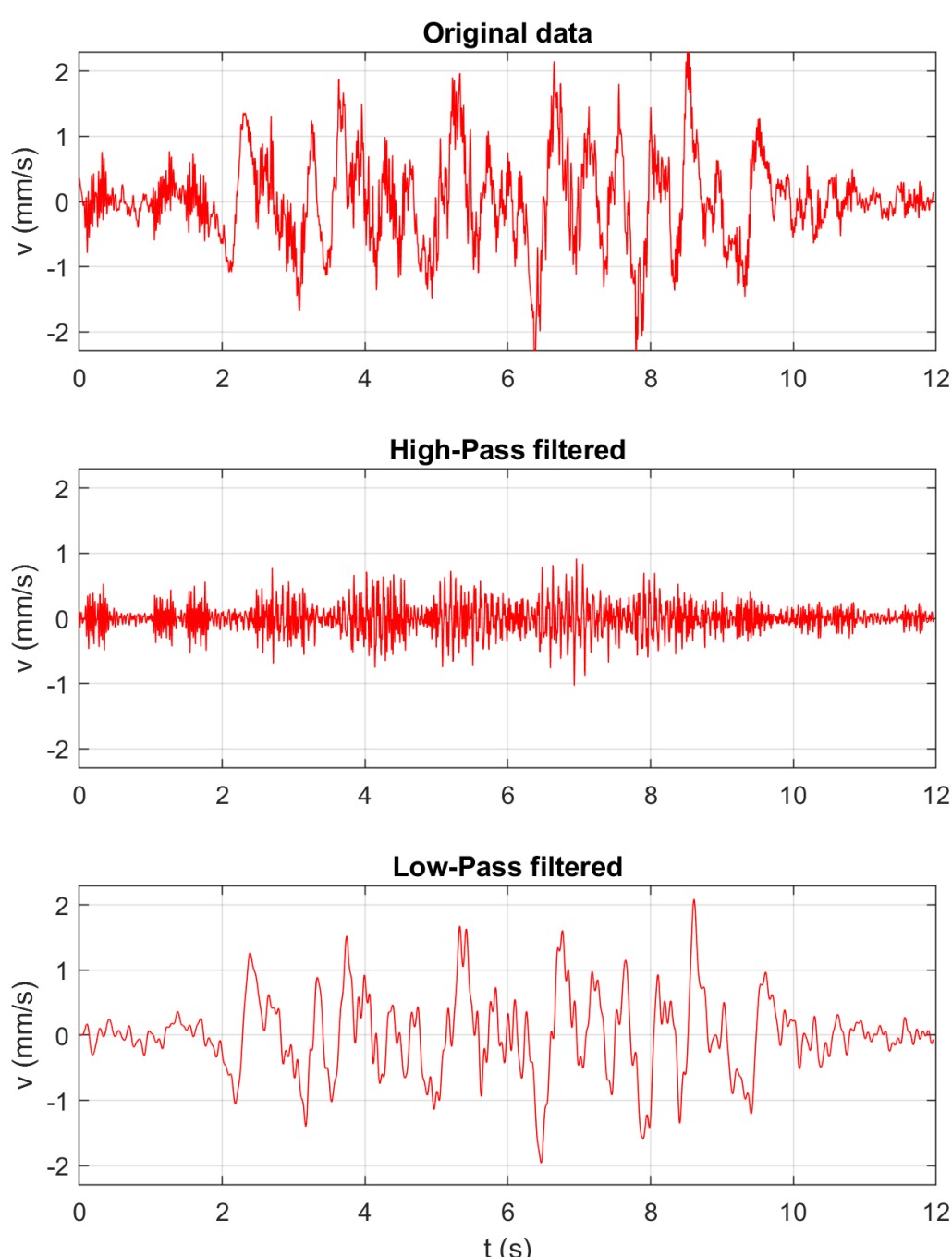

**Figure 13.** Time history, high-pass, and low-pass frequency filtered signal for node A2 during train is present.

Figure 14 shows the numerical results for velocities of the same node after the train left around 10 s. It also illustrates FE data under the high-frequency filter and low-frequency filter. A free decay process is simulated in this part. Then we do the Fourier transformation to the response during the free decay time period and compare the first-mode frequency with the Eigen analysis for the structure, as demonstrated in Figure 15.

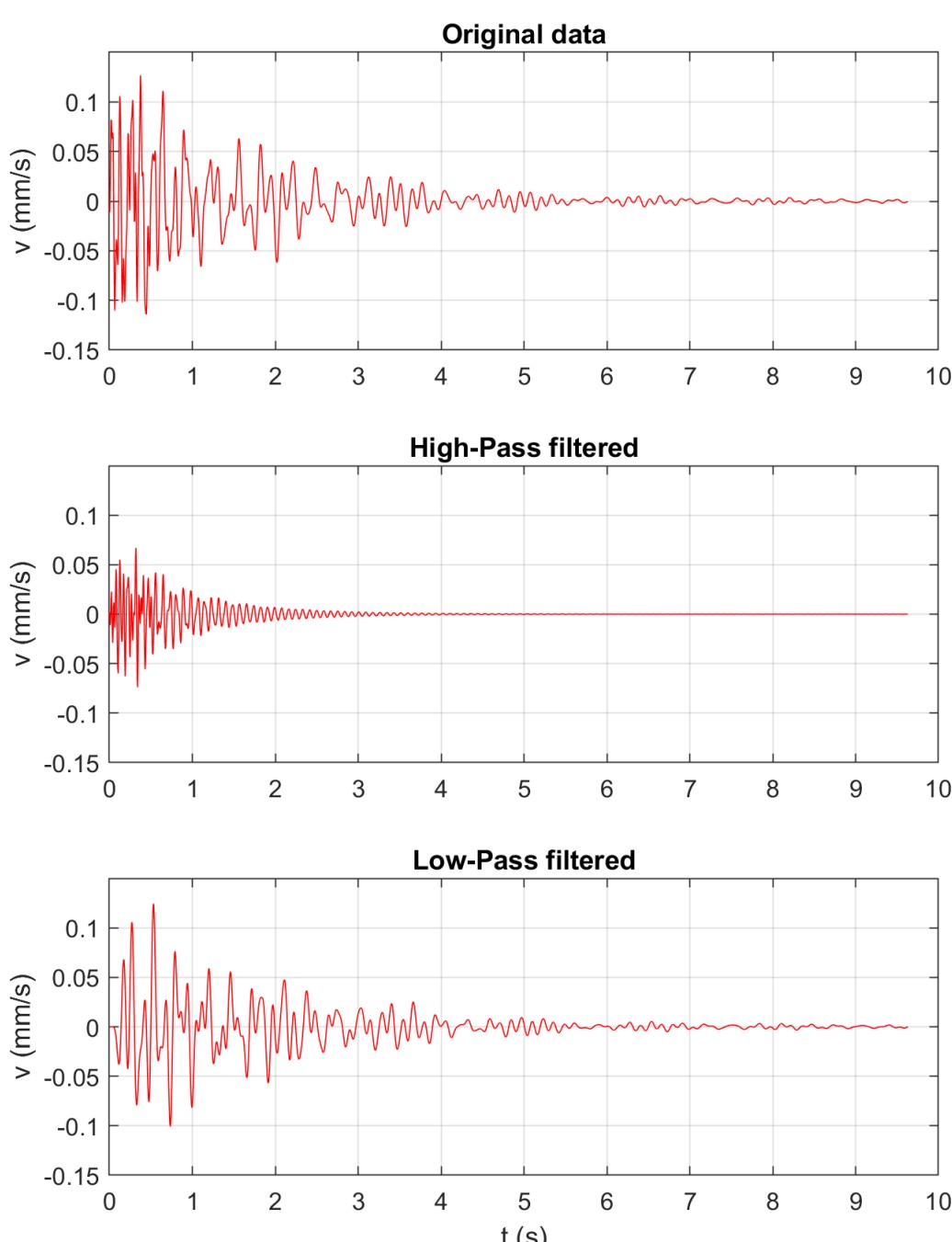

**Figure 14.** Time history, high-pass, and low-pass frequency filtered signal for node A2 free decay.

The results show a good correlation, indicating the structure's characteristics are captured well. The contribution of the vehicle interprets slight discrepancies.

Noticeably, field test data have been omitted from our study due to concerns regarding environmental effects. In reality, the entire structure encompasses more than three spans, spanning over 1000 m, and exists within a complex surrounding environment. The field test data, gathered from dispersed sensors, is significantly influenced by environmental factors such as noise and pedestrian activities. Consequently, this data cannot be considered a reliable reference to validate our approach, given that our finite-element model focuses on an idealized environment with only three spans.

Despite the constraints on the availability of field test data, our research still delivers valuable contributions to the field of high-speed rail (HSR). We remain committed to continuous improvement, and we pledge to incorporate field test data in future studies,

provided it becomes accessible and reliable. By doing so, we aim to enhance the robustness and applicability of our findings to real-world scenarios.

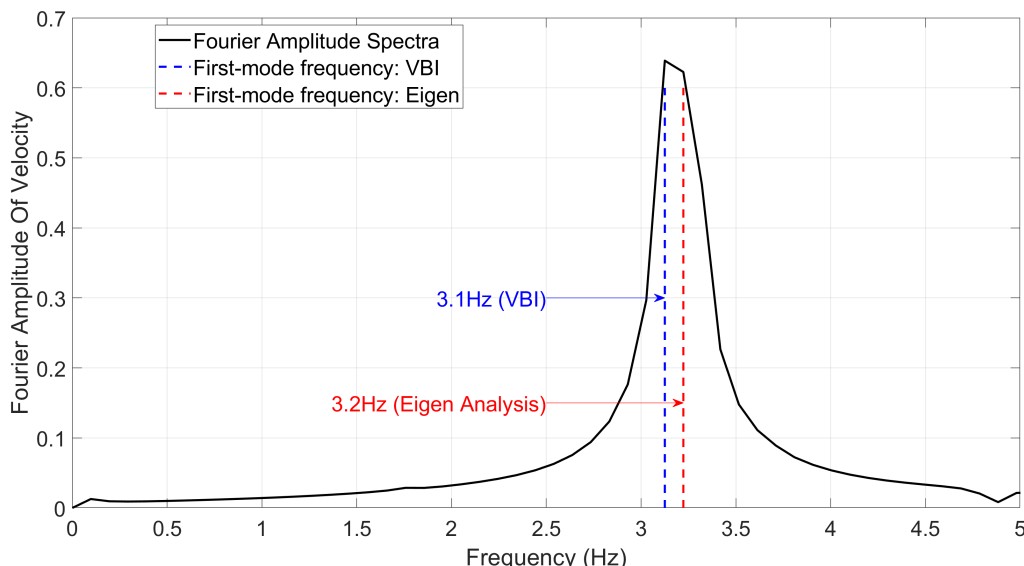

**Figure 15.** FFT of the node A2 for free decay.

## 5. Conclusions and Discussions

In this paper, we have implemented the vehicle–bridge interaction (VBI) element theory proposed by Yang [43] in ABAQUS using a user-element subroutine (UEL). To verify the accuracy and reliability of the implementation, we conducted verification studies involving simple VBI problems. These problems include a moving sprung mass acting on a simply supported beam and a two-wheel suspended rigid beam moving on a simply supported beam system. The obtained results for the displacement and velocity at the midpoint of the beam, using the VBI element theory, exhibit a close agreement with the analytical solution based on the first-mode approximation obtained through MATLAB. This successful match between the analytical solution and the results obtained from the implementation in ABAQUS demonstrates the effectiveness and validity of the implemented VBI element theory in ABAQUS.

Furthermore, we showcased a practical application example to demonstrate the implementation's real-world relevance. For this purpose, we developed a detailed 3D model to perform an FE analysis of the vehicle–bridge interaction on a multi-girder high-speed railway bridge. In the model, solid elements were used to represent the bridge deck and piers, ensuring a comprehensive representation of their structural behavior. On the other hand, beam elements were employed to model the rails, capturing their dynamic response under vehicle loads effectively. To simulate the connection between the rail and rail pad, we implemented spring and dashpot systems, allowing for an accurate representation of the dynamic interaction between the rail and the rest of the structure during train movement.

A good correlation is demonstrated between the Eigen analysis and FE simulation for the train-left time window. Partial validation is illustrated in comparison to the natural frequency of the bridge. An acceptable discrepancy is interpreted by the vehicle load effect and road roughness contribution. Based on Yang's theory, the interaction between wheels and rail is derived through equations of motion for the vehicle and rail parts. Consequently, an iteration process within a single time step is not required. Thus, the implemented procedure for the VBI element theory in ABAQUS using the user element (UEL) is found to be both applicable and convenient. This successful implementation opens the door for future studies on vehicle–bridge interactions, allowing for more in-depth and comprehensive analyses of such dynamic interactions in high-speed railway systems and other relevant scenarios. The versatility and reliability of the implemented approach make it a valuable tool for exploring and understanding the complexities of

vehicle–bridge interactions, paving the way for further advancements, which enables researchers to conduct broader research on:

- The effects of culvert properties (its depth from track, stiffness, and geometry) on high-speed train dynamics;
- The impacts of embankment on high-speed train dynamics;
- The investigation of Rayleigh wave propagation in soil under the trainload;
- The exploration of the bridge abutment transition zones and their optimal design;
- The investigation of the soil boundary conditions such as perfect-matched layer;
- The analysis of derailment risk under moderate-to-large earthquake excitation on both regular track and long-span bridges;
- The investigation of the influence of sub-grade and soil nonlinearities on high-speed train dynamics and critical speed analyses.

**Supplementary Materials:** The following supporting information can be downloaded at: https://www.mdpi.com/article/10.3390/app13158812/s1, File S1: SM2023.m; File S2: uel_VBIBPS3D.for; File S3: MRB2023.m; File S4: UEL3Dbeam8w.for; File S5: uel_VBIBPS3D2019111116w.for.

**Author Contributions:** Conceptualization, Y.D. and L.S.; methodology, L.S.; software, Y.D.; validation, Y.D. and L.S.; formal analysis, Y.D.; investigation, L.S.; resources, L.S.; data curation, L.S.; writing—original draft preparation, Y.D.; writing—review and editing, W.Z. and E.T.; visualization, W.Z., E.T. and A.S.; supervision, E.T.; project administration, L.S.; funding acquisition, E.T. All authors have read and agreed to the published version of the manuscript.

**Funding:** This research was funded by the National Natural Science Foundation of China (Grant No. 51608482).

**Institutional Review Board Statement:** Not applicable.

**Informed Consent Statement:** Not applicable.

**Data Availability Statement:** Not applicable.

**Acknowledgments:** The study conducted in this paper is supported by the University of California, Los Angeles. The authors would like to express their appreciation for this generous support. The opinions and views presented in this paper are those of the authors. Thanks to Junwei Chen, a student of the Civil Engineering Department at the Zhejiang University of Technology, for his commitment, advice, and technical support during the simulation process.

**Conflicts of Interest:** The authors declare no conflict of interest.

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
