# Peer review of "A Vehicle–Bridge Interaction Element: Implementation in ABAQUS and Verification"

_applsci, doi:10.3390/app13158812_

Round 1

Reviewer 1 Report

The comments of the reviewer on applsci-2514498-peer-review-v1:

In this paper, the element theory of vehicle-bridge-interaction (VBI) presented by Yang is implemented in ABAQUS through a user element subroutine (UEL). The authors performed verification studies based on simple VBI problems, including a moving mass sprung resting on a simply supported beam and a two-wheeled rigid rod moving on a simply supported beam system. The displacement and velocity results for the midpoint of the beam using the VBI element theory demonstrate a good agreement with the analytical solution based on the first-state approximation obtained by MATLAB realization, which shows the availability of the implementation for the VBI analysis via ABAQUS. A brief review of this article reveals that it is a fairly well-written paper, and the results have been appropriately tabulated and graphically presented; however, there are major/serious issues that require further attention of the authors before it can be recommended for publication:

  1. What are the original aspects of this paper? At present, there are many works on the dynamic response of bridge structures under moving trains and vehicles and with regard to the high-speed computers, their full analyses and interactions can be readily carried out. Therefore, the need for the specific approach developed in this paper is questionable, requiring more detailed explanations from the authors. 

2.     According to the used shape functions used in modeling the 2-node beam element, the shape functions have been constructed based on the Euler-Bernoulli beam theory. However, we cannot employ this theory for vibrational analyses of all problems associated with VBIs, particularly when the role of shear deformation becomes important. Can the authors display this crucial issue what are the practical limitations of the proposed numerical approach based on the shape functions given in Eqs. (8)-(12)? 

3.     Under the umbrella of comment#1, when we should use shear deformable beam theories in the VBI analysis? The following reference works that deal with the role of shear deformation on the vibrations of beam-like structures subjected to moving loads and masses should be cited and explained: 

(*) https://doi.org/10.1016/S0003-682X(97)00097-2

(*) https://doi.org/10.1115/1.3147165 

(*) https://doi.org/10.1016/S0965-9978(02)00003-0 

(*) https://doi.org/10.1016/j.jsv.2008.08.010 

(*) https://doi.org/10.1243/09544062JMES119 

(*) https://doi.org/10.1016/j.euromechsol.2011.07.008 

(*) https://doi.org/10.1016/j.ijmecsci.2019.01.033

(*) https://doi.org/10.1016/j.compstruct.2014.03.045   

4.     The presented various lines in Figure 13 are somehow untidy, making an unreadable figure. Hence, the authors are invited to appropriately replot them, arriving at a more vivid figure. 

5.     The fundamental phenomena in the VBI analysis have been NOT elucidated at all. For example, the possible existence of the resonance state was not discussed. In addition, the concept of critical velocity has been not explained. To this end, the following papers that display these crucial factors, even at the nanoscale through developing more continuum mechanics such as the nonlocal elasticity theory, should be mentioned and cited in the paper: (*) Assessment of nanotube structures under a moving nanoparticle using nonlocal beam theories. Journal of Sound and Vibration. 2010;329(11):2241-64; (*) Application of nonlocal beam models to double-walled carbon nanotubes under a moving nanoparticle. Part I: theoretical formulations. Acta Mechanica. 2011;216:165-95; (*) Application of nonlocal beam models to double-walled carbon nanotubes under a moving nanoparticle. Part II: parametric study. Acta Mechanica. 2011;216(1-4):197-206.

6.     The English of the paper needs further attention from the authors. In the present version, there exist many grammatical errors, typos, and long sentences throughout the paper’s manuscript that should be carefully and suitably revised. 

7. Minor issues:

7.1. It is not needed that the abbreviation of the Perfect-Matched-Layer be given at the end of the article since it has been not used again. 

7.2. In all presented graphs, the x-label and y-label expressions like “Time” and “Velocity” should be replaced by their specific math form (using “t” and “v”) for the sake of more consistency with the existing literature.

7.3. In the caption of Figure 1, “flow” should be modified to “flowchart”.

7.4. In the caption of Figure 2, “Scheme” should be revised to “Schematic representation”. 

7.5. In the caption of Figure 4, “the bar system” should be replaced with a more technical expression since “bar” is NOT an appropriate word for the demonstrated system. 

7.6. In Figure 1, the presented vector factors du, dw, and dc have been not defined.

Minor editing of English language required:

==============================

It is not needed that the abbreviation of the Perfect-Matched-Layer be given at the end of the article since it has been not used again.

In all presented graphs, the x-label and y-label expressions like “Time” and “Velocity” should be replaced by their specific math form (using “t” and “v”) for the sake of more consistency with the existing literature.

In the caption of Figure 1, “flow” should be modified to “flowchart”.

In the caption of Figure 2, “Scheme” should be revised to “Schematic representation”.

In the caption of Figure 4, “the bar system” should be replaced with a more technical expression since “bar” is NOT an appropriate word for the demonstrated system.

In Figure 1, the presented vector factors du, dw, and dc have been not defined.

Reviewer 2 Report

This paper presented a ABAQUS implementation of vehicle-bridge interaction (VBI) element theory and verified the implementation using two analytical solutions and one case study. The paper is well-written and the only issue is that in the title, "Implementation to ABAQUS" should be better replaced by "Implementation in ABAQUS". Apart from this, this paper is recommended to publish

Author Response

Response to Reviewer 2 Comments

Point 1: This paper presented a ABAQUS implementation of vehicle-bridge interaction (VBI) element theory and verified the implementation using two analytical solutions and one case study. The paper is well-written and the only issue is that in the title, "Implementation to ABAQUS" should be better replaced by "Implementation in ABAQUS". Apart from this, this paper is recommended to publish.

Response 1: Dear Reviewer, thank you so much for your comments and affirmation. I have updated the title to "Implementation in ABAQUS". It will be shown in the revised manuscript after submission.

Reviewer 3 Report

1)     It is recommended to include some statistical results in the abstract to indicate the performance of the proposed interaction element to address the vehicle-bridge contact.

2)     I am not sure if high-speed rail can be the context of this study, as vehicle-bridge interaction has a more wide application scope.

3)     The contact problem in railway dynamic systems should be extensively summarised in the literature review due to the complexity of this system as reported in [1-2].

[1] Song, Y., Wang, Z., Liu, Z., & Wang, R. (2021). A spatial coupling model to study dynamic performance of pantograph-catenary with vehicle-track excitation. Mechanical Systems and Signal Processing, 151, 107336. https://doi.org/10.1016/j.ymssp.2020.107336

[2] Carnicero, A., Jimenez-Octavio, J. R., Sanchez-Rebollo, C., Ramos, A., & Such, M. (2012). Influence of track irregularities in the catenary-pantograph dynamic interaction. Journal of Computational and Nonlinear Dynamics, 7(4), 041015. https://doi.org/10.1115/1.4006735

4)     In the context of railway engineering, there is a complex track layer between the bridge and the vehicle. How to model the details of the track using the present method?

5)     One of the drawbacks is that the present method has not been verified by comparison with other numerical methods or field test data. Please comment on this issue.

Round 2

Reviewer 1 Report

The revised paper can be accepted.

Reviewer 3 Report

Thanks for good revisions. It deserves publication.